# IS MY "RED" YOUR "RED"?: UNSUPERVISED ALIGNMENT OF QUALIA STRUCTURES VIA OPTIMAL TRANSPORT

**Genji Kawakita**[*]
Imperial College London
{g.kawakita22}@imperil.ac.uk

**Ariel Zeleznikow-Johnston**[*]
Monash University
{ariel.zeleznikow-johnston}@monash.edu

**Ken Takeda**[*]
The University of Tokyo
{tkkentakeda1248}@g.ecc.u-tokyo.ac.jp

**Naotsugu Tsuchiya**[†]
Monash University
{naotsugu.tsuchiya}@monash.edu

**Masafumi Oizumi**[†]
The University of Tokyo {c-oizumi}@g.ecc.u-tokyo.ac.jp

## ABSTRACT

Whether one person's experience of "red" is equivalent to another's has long been considered unanswerable. One promising approach to resolving this fundamental question about consciousness is the intersubjective comparison of the similarity relations of sensory experiences, termed "qualia structures". Conventional methods for comparing similarity relations largely sidestep the issue, assuming that experiences elicited by the same stimuli are matched across individuals, and thus ruling out the possibility that my "red" could be your "blue". Here, we present an unsupervised optimal transport method for assessing the similarity of qualia structures without presupposing correspondences between individuals. To validate and demonstrate the utility of the proposed approach, we analyzed a massive dataset of subjective color similarity judgments from color-neurotypical and color-blind participants. We show that optimal correspondences between qualia structures within color-neurotypical participants can be "correctly" aligned based solely on similarity relations. In contrast, qualia structures from color-blind individuals could not be aligned with those of color-neurotypicals. Our results offer quantitative evidence for the interindividual structural equivalence or difference of color qualia, implying that a color-neurotypical person's "red" is indeed another color-neurotypical's "red", but not a color-blind person's "red", from a structural perspective. This method is applicable across modalities, enabling general structural exploration of subjective experiences.

## 1 INTRODUCTION

The question of whether sensory experiences are intersubjectively equivalent is a central concern in the study of consciousness. Some researchers consider the question impossible to answer because of the intrinsic, ineffable, and private nature of subjective experience (1). Although direct description of our experiences in a fashion that allows for intersubjective comparison may be impossible, indirect characterization of experience is empirically feasible and is considered a promising research

---

[*]These authors contributed equally to this work.
[†]These authors contributed equally to this work.

program (6; 7; 2; 8; 9; 4; 10; 3; 11; 12; 5). One notable approach is to analyze reports of subjective similarities between sensory experiences (13; 14; 15; 16; 17). Relationships between sensory experiences, such as similarity, allow for the structural investigation of phenomenal consciousness (18; 19; 20).

Based on this idea, we formally introduce a new paradigm, which we call the "qualia structure paradigm" (Fig. 1a). This paradigm consists of two main steps. The first step is to collect detailed subjective reports about the relations between sensory experiences (qualia) through psychophysics experiments (17). We then estimate the embeddings of qualia for different participants that best explain the participants' similarity judgements. The set of qualia embeddings is represented as points in space (Fig. 1a) and is considered as a 'qualia structure'. Importantly, the relations of the qualia are represented as the "distances" of the embeddings, and we can estimate the dissimilarity matrices of the qualia based on the distance (e.g., Euclidean distance) between the embeddings.

Having obtained two qualia structures from different participants (Fig. 1a), the second step is to compare these structures and evaluate the extent to which they are similar, without assuming a correspondence between individual qualia from one structure to the other. This is in contrast to previous analyses of two qualia structures or dissimilarity matrices, which typically assumes an "external" correspondence at the stimulus level: my experience of "red" evoked by a red stimulus corresponds to your experience of "red"(Fig. 1b). This type of supervised comparison between dissimilarity matrices, known as Representational Similarity Analysis (RSA), has been widely used in neuroscience to compare various similarity matrices obtained from behavioral and neural data (21; 22). However, there is no guarantee that the same stimulus will necessarily evoke the same corresponding subjective experience in different individuals. Accordingly, when considering which stimuli evoke which qualia for different individuals, we need to consider all possibilities of correspondence. For example, my "red" could correspond to your "red", "green", "purple", or it could lie somewhere between your "orange" and "pink" (Fig. 1c).

## 1.1 UNSUPERVISED ALIGNMENT OF QUALIA STRUCTURES

To account for all possible correspondences, we propose to use an unsupervised alignment method for quantifying the degree of similarity between qualia structures. As shown in Fig. 1d, in unsupervised alignment, we do not attach any external (stimuli) labels to the qualia embeddings. Instead, we try to find the best matching between qualia structures based only on their internal relationships (see Methods). After finding the optimal alignment, we can use external labels, such as the identity of a color stimulus (Fig. 1e), to evaluate how the embeddings of different individuals relate to each other. This allows us to determine which color embeddings correspond to the same color embeddings across individuals, and which do not. Checking the assumption that these external labels are consistent across individuals allows us to assess the degree of inter-individual correspondences between qualia structures obtained from different participants.

To this end, we used the Gromov-Wasserstein optimal transport (GWOT) method (23), which has been applied with great success in various fields (25; 26; 27; 28; 24; 29)). GWOT aims to find the optimal transportation plan $\Gamma$ between two point clouds in different domains based on the distance $D$ (or $D'$) between points within each domain (Fig. 2). Importantly, the distances (or correspondences) between points "across" different domains are not given while those "within" the same domain are given. GWOT aligns the point clouds according to the principle that a point in one domain should correspond to a point in the other domain that has a similar relationship to other points within its domain (see Methods for the details). The optimal transportation plan $\Gamma$ can be interpreted as the probability of an embedding in one qualia structure corresponding to an embedding in the other qualia structure. By using the optimal transportation plan $\Gamma$, we can evaluate the degree to which two qualia structures match correctly.

To assess the validity and utility of the qualia structure paradigm, we apply this framework to a similarity judgment data-set involving 93 colors as a representative and tractable case study. The relatively large number of colors enables us to investigate complex and nuanced qualia structures of colors, which is less feasible with previous datasets examining a smaller number of colors (13; 15; 16; 17). In addition, the large number of colors necessitates the computational efficiency of our method, as a brute-force search method considering all possible correspondences used in previous studies (e.g. (30)) would not be practical.

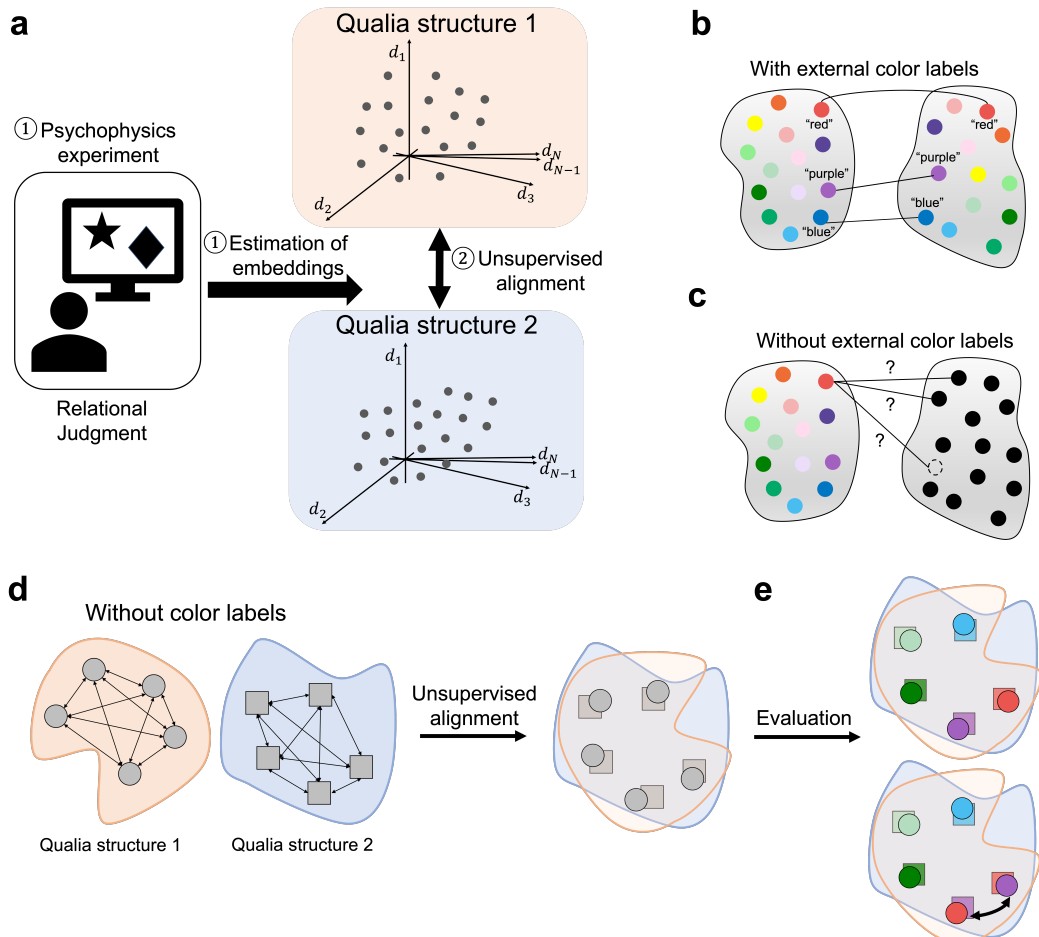

Figure 1: **Schematics of concepts in the qualia structure paradigm.** (a) Two steps in the qualia structure paradigm. The first step is to collect subjective reports through relational judgments between stimuli that enable estimation of the relational structure of sensory experiences, i.e., qualia structure. The second step is to align qualia structures from different individuals in an unsupervised manner to quantify the degree of similarity of their qualia structures. (b) Supervised alignment of color qualia structures, which assumes correspondence between qualia evoked by the same external stimuli across different individuals. (c) Unsupervised alignment of color qualia structures, which does not assume correspondence between qualia across different individuals. All possible correspondences are taken into consideration. A particular color quale for an individual may not have an exact correspondence to a particular quale of another individual, as indicated by the dotted circle. (d) Aligning qualia structures of different individuals in an unsupervised manner without any external labels, solely based on the internal relationships of the embeddings. (e) Evaluation of unsupervised alignment using external labels.

In this study, using data from both color-neurotypical participants and color-atypical participants, we address two questions: (1) whether color qualia structures can be aligned within color-neurotypical and color-atypical participant groups, separately, and (2) whether color qualia structures can be aligned across color-neurotypical and color-atypical participant groups. The first analysis is necessary to determine whether there are structures that are sufficiently common across participants to be alignable within color-neurotypical participants or within color-atypical participants. These cases also serve as a positive control, where we should be able to align two qualia structures, using our unsupervised alignment method, which relies only on the internal relationships. After establishing the validity of our methods, we quantified the degree to which the qualia structures of these two populations can be aligned.

Qualia structure 1     Qualia structure 2

$$\text{GWD} = \min_{\Gamma} \sum_{i,j,k,l} \|D_{ij} - D'_{kl}\|^2 \Gamma_{ik} \Gamma_{jl}$$

Figure 2: **Schematic of Gromov-Wasssersetin optimal transport.** The elements of matrices $D$ and $D'$ are the distances between the embeddings. $\Gamma$ is the transportation matrix indicating the probability of an embedding in one qualia structure corresponding to an embedding in the other qualia structure.

## 2   RESULTS

### 2.1   MASSIVE ONLINE EXPERIMENT OF COLOR SIMILARITY JUDGEMENT

We collected similarity judgments between 93 colors from 426 color-neurotypical and 257 color-atypical participants using an online cloud sourcing service. 257 color-atypical participants self-reported as color blind. Their reports were verified by a modified online Ishihara test (see Methods). Each participant provided pairwise dissimilarity judgments for a randomly assigned subset of the 4,371 possible color pairs (including the same color pairs).

In this study, we considered the alignment between the color similarity structures on a participant group basis by aggregating the similarity judgments of many participants to estimate a group-level color similarity structure. This is because the number of color pairs reported by each participant was only 162 at most, which is too small to reliably estimate the entire color similarity structure of 93 colors. As described below, we first considered alignment within color-neurotypical groups, then within color-atypical groups, and then finally between these participant groups.

### 2.2   UNSUPERVISED ALIGNMENT OF COLOR QUALIA STRUCTURES

#### 2.2.1   UNSUPERVISED ALIGNMENT BETWEEN COLOR-NEUROTYPICAL PARTICIPANTS

First, we considered the alignment within color-neurotypical participants (between subgroups of color-neurotypical participants). We aggregated the similarity judgment responses of 128 randomly selected color-neurotypical participants out of the total 426 color-neurotypical participants and created a pair of non-overlapping participant groups, each consisting of 128 participants. We repeated this random sampling 20 times. We show the results of one of the 20 samples in Figs. 3a-e, and all the results of the 20 different samples in Fig. 3f.

As a demonstration, we show the embeddings of 93 colors for a certain random pair of groups, denoted as Group T1 and T2 in Fig. 3a. For each group, we estimated the embeddings that best explained the experimentally obtained similarity responses, based on the procedure described in detail in Methods. We then applied principal component analysis (PCA) to reduce the dimensions of the embeddings to 3 for visualization (Fig. 3a). From the estimated embeddings, we obtained the dissimilarity matrices $D$ by computing the Euclidian distances between the estimated embeddings (Fig. 3b), where the entry, $D_{ij}$, represents the subjective dissimilarity between the two experiences of the $i$-th and $j$-th colors.

We then compared the two qualia structures by performing an unsupervised alignment based on entropic GWOT (Eq. 10) on the estimated dissimilarity matrices (Fig. 3b). Since entropic GWOT is a non-convex optimization problem involving hyperparameter search of $\epsilon$, which controls the degree of entropy regularization, we performed a total of 200 optimization iterations with different $\epsilon$ values

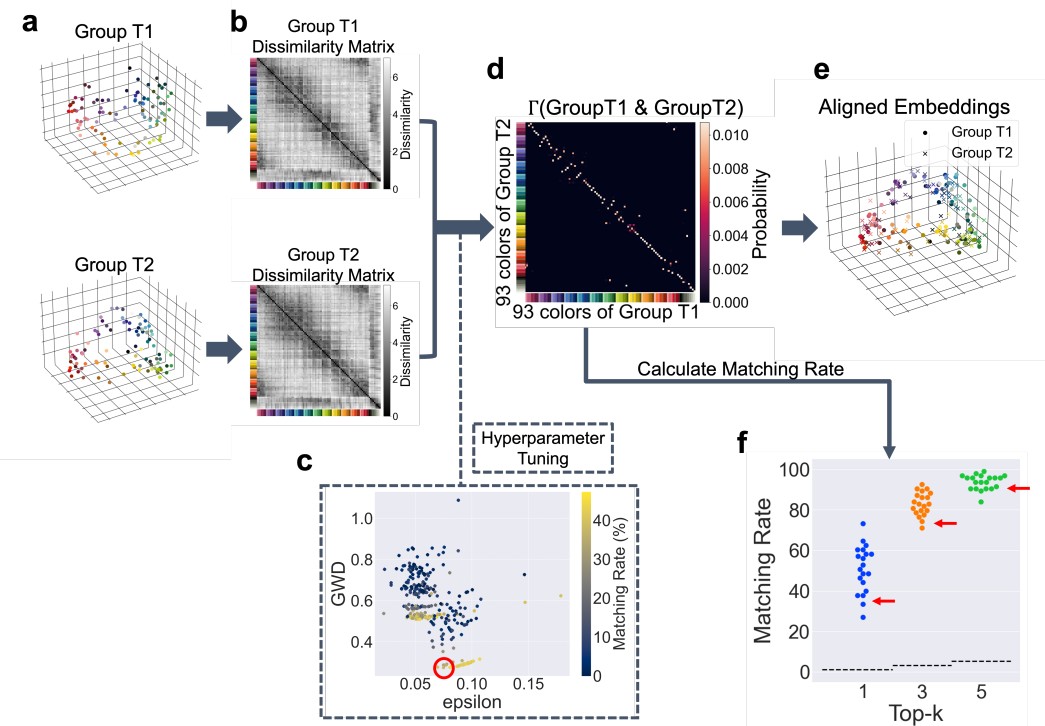

Figure 3: **Unsupervised alignment between qualia structures of color-neurotypical partici-pants groups** (a) Estimated embeddings of 93 colors from two color-neurotypical participant groups (Group T1 and T2). (b) Dissimilarity matrices of 93 colors from Group T1 and T2 obtained from the embeddings. (c) The optimization results over 200 iterations with different $\epsilon$ values. GWD values of local minima represented by points are shown with respect to $\epsilon$. Colors represent the top-1 matching rate of unsupervised alignment. (d) Optimal transportation plan $\Gamma$ between the dissimilarity matrices of Group T1 and T2. (e) Aligned embeddings of Group T1 and T2 plotted in the embedded space of Group T1. (f) The top-k matching rate of unsupervised alignment for 20 random pairs of participant groups. The chance levels are indicated by the dotted lines.

and initializations of transportation plans to search for a global optimum. The points in Fig. 3c correspond to the local minimum found in each iteration of the optimization performed on different $\epsilon$ values. We selected the local minimum with the lowest GWD as the optimal solution (shown in the red circle in Fig. 3c).

From the optimization process, we finally obtained the optimal transportation plan $\Gamma$ between Group T1 and T2 (Fig. 3d). As shown in Fig. 3d, most of the diagonal elements in $\Gamma$ have high values, indicating that most of the colors in one group correspond with a high probability to the same colors in the other group. To quantitatively assess the degree of correspondence, we computed the top-1 matching rate of the 93 colors (see Methods for details), which was 38 %. As can be seen in Fig. 3c, the local minima with low GWD (in the y-axis) tend to yield a high matching rate (points with yellowish color), which is necessary for unsupervised alignment to achieve a high matching rate.

After applying GWOT, we performed an alignment of the two sets of embeddings, which is visual-ized in Fig. 3e. Although the optimized transportation plan $\Gamma$ provides the rough correspondence between the embeddings of the qualia structures, we can find a more detailed mapping in the orig-inal space of the embeddings. As described in Methods, we aligned the embeddings of Group T1 (denoted by $X$) with those of Group T2 (denoted by $Y$) by finding the orthonormal rotation matrix $Q$ using the optimized transportation plan $\Gamma$ obtained by GWOT. In Fig. 3e, we plotted the embed-dings of group T1, $X$, and the aligned embeddings of group N2, $QY$. This visualization clearly demonstrates that the embeddings of similar colors from both groups are closely located to each other, indicating that similar colors are "correctly" aligned by the unsupervised alignment. Note that

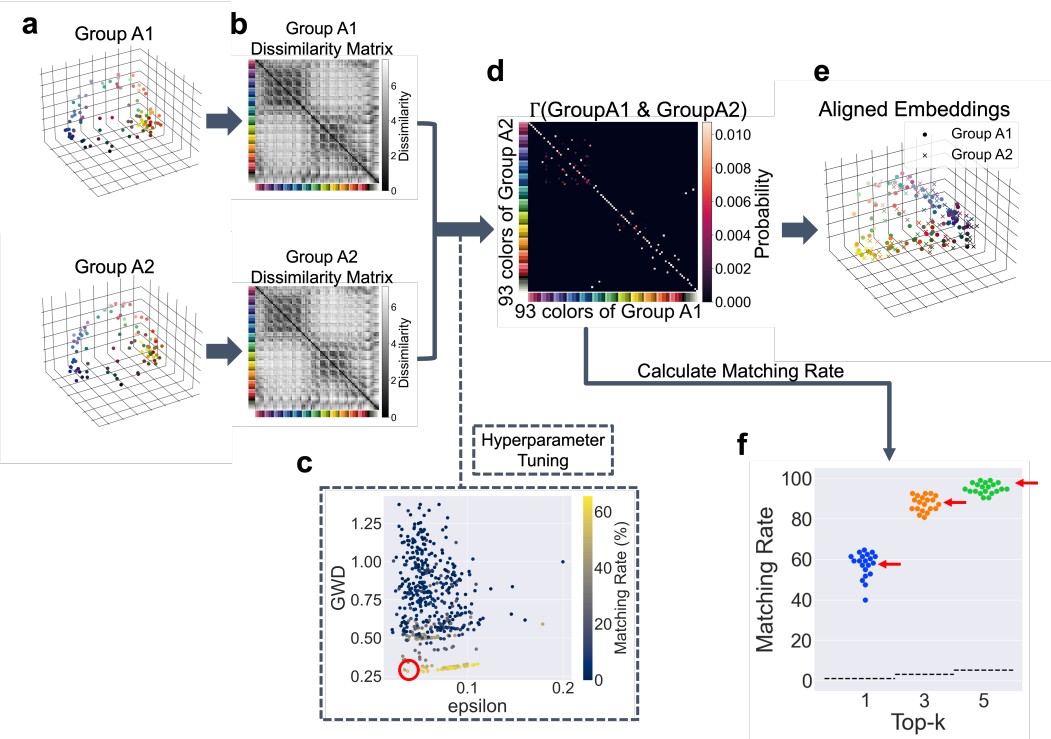

Figure 4: **Unsupervised alignment between qualia structures of color-atypical participants groups** (a) Estimated embeddings of 93 colors from two color-atypical participant groups (Group A1 and A2). (b) Dissimilarity matrices of 93 colors from Group A1 and A2 obtained from the embeddings. (c) The optimization results over 200 iterations with different $\epsilon$ values (x-axis). GWD values (y-axis) of local minima represented by points are shown with respect to $\epsilon$. Colors represent the top-1 matching rate of unsupervised alignment. (d) Optimal transportation plan $\Gamma$ between the dissimilarity matrices of Group A1 and A2. (e) Aligned embeddings of Group A1 and A2 plotted in the embedded space of Group A1. (f) The top-k matching rate of unsupervised alignment for 20 random pairs of participant groups. The red arrows indicate the matching rates for the example shown in a-e. The dotted lines indicate the chance levels.

although the colors in Fig. 3e are used for evaluation purposes only, the entire alignment process was performed in a purely unsupervised manner, without relying on the color labels.

By performing the same analysis for all 20 random pairs of participant groups, we obtained the top-k matching rate of the 20 random samples (Fig. 3f). The results of the particular example shown in Fig. 3d are highlighted by the red arrows. The averages of the top 1, 3, and 5 matching rate over 20 random samples are 51%, 83%, and 94%, respectively. Importantly, all values of the matching rates from 20 random samples are well above the respective chance level (1.1%, 3.2%, 5.4%). This suggests that there are sufficiently common structures among color-neurotypical participants to enable their alignment in an unsupervised manner. This result validates the effectiveness of our unsupervised alignment method based on GWOT, which relies solely on the internal relationships of color similarity structures, and demonstrates its effectiveness in scenarios where it is expected to work.

### 2.2.2 UNSUPERVISED ALIGNMENT BETWEEN COLOR-ATYPICAL PARTICIPANTS

Next, we considered the alignment between different color-atypical participant groups (see Methods and Supplementary Fig. S1 for screening details). Most of these participants are likely to have red-green color vision deficiencies , as further detailed in Supplementary Fig.1.

We investigated whether the qualia structures of color-atypical participant subgroups could be aligned similarly to those of color-neurotypical participants. To this end, we replicated the analysis performed with color-neurotypical participants, in which pairs of participant groups consisting of randomly selected 128 participants were formed 20 times. The results for one particular pair of color-atypical groups, Group A1 and A2, are presented in several figures: Fig. 4a shows the embeddings, Fig. 4b shows the dissimilarity matrices , and Fig. 4c details the GWOT optimization results over 200 iterations, all in a format similar to the previous analysis. Fig. 4d shows the optimal transportation plan $\Gamma$ for Group A1 and A2. We can see that most of the diagonal elements in $\Gamma$ have high values, resulting in a top-1 matching rate of 59%. In Fig. 4e, the embeddings of group A1 and the aligned embeddings of group A2 are plotted, demonstrating the effectiveness of the unsupervised alignment, as evidenced by the close placement of similar colors from both groups.

The top-k matching rate of the 20 random samples (Fig. 4f) further confirms the validity of our methods. The average of the top 1, 3, and 5 matching rate over 20 random samples is 57%, 87%, and 95%, respectively. All values of the matching rates significantly exceed their corresponding chance levels, which are 1.1%, 3.2%, and 5.4%. This result suggests that at the group level, there are sufficient common structures among color-atypical participants that they can be aligned in an unsupervised manner, even though the degree of red-green color deficiency varies among individual participants.

### 2.2.3 UNSUPERVISED ALIGNMENT BETWEEN COLOR-NEUROTYPICAL AND COLOR-ATYPICAL PARTICIPANTS

Finally, we investigated to what extent the similarity structures of color-neurotypical and color-atypical participants could be aligned. For this purpose, we separately sampled 128 participants from both color-neurotypical and color-atypical participants and paired a color-neurotypical participant group with a color-atypical participant group. This procedure was repeated 20 times, resulting in 20 pairs, each consisting of a group of the color-neurotypical participants and a group of the color-atypical participants. As an illustrative case among these 20 random pairs, we show the estimated embeddings (Fig. 5a) and the dissimilarity matrices of a color-neurotypical and a color-atypical participant group (Fig. 5b), labeled as Group T1 and A1, respectively. Upon visual inspection of the embeddings of the color-neurotypical and atypical group in Fig. 5a, we can see that while the overall structures of the color-neurotypical and atypical group are similar, there are distinct differences. In particular, greenish and reddish colors are close in the embedding space of the color-atypical participants, as highlighted by the red circle in Fig. 5a, while they appear distant in the space of the color-neurotypical participants. Despite these differences, the two dissimilarity matrices in Fig. 5b show a significant degree of similarity. This is quantitatively supported by the Pearson's correlation coefficient of 0.66 between the dissimilarity matrices of Group T1 and Group A1. While this coefficient is somewhat lower than the Pearson's correlation coefficient between the dissimilarity matrices of the color-neurotypical groups (Group T1 and T2 in Fig. 3b, with $\rho = 0.88$), and between the color-atypical groups (Group A1 and A2 in Fig. 4b, with $\rho = 0.91$), it still represents a substantial correlation.

Using the dissimilarity matrices, we performed the unsupervised alignment based on GWOT between the color-neurotypical participant group and the color-atypical participant group. We performed a total of 200 optimization iterations on different $\epsilon$ values and selected the local minimum with the lowest GWD as the optimal solution (highlighted by the red circle in Fig. 5c). In Fig. 5c, we observe that local minima with low values of GWD have low matching rates, leading to unsuccessful alignment. As can be seen in Fig. 5d, the optimal transportation plan $\Gamma$ is not lined up diagonally (the diagonal elements of $\Gamma$ are very small), unlike the optimal transportation plan between the color-neurotypical participant groups shown in Fig. 3d or that between the color-atypical participant groups shown in Fig. 4d. The optimal solution with the lowest GWD has a top-1 matching rate of 1.1%, which is close to the chance level (1.1%). In Fig. 5e, we plotted the embeddings of Group T1 and the aligned embeddings of Group A1. Unlike the results seen in Fig. 3e and in Fig. 4e, here the embeddings of similar colors from the two groups are not positioned closely, indicating that similar colors are not correctly aligned by the unsupervised alignment.

By performing the same analysis for all 20 random pairs of the participant group, we obtained the top-k matching rate of the 20 random samples (Fig. 5f). The averages of the top 1, 3, and 5 matching rates over 20 random samples are 3.8%, 8.3%, 11.7%, respectively, which are slightly higher but

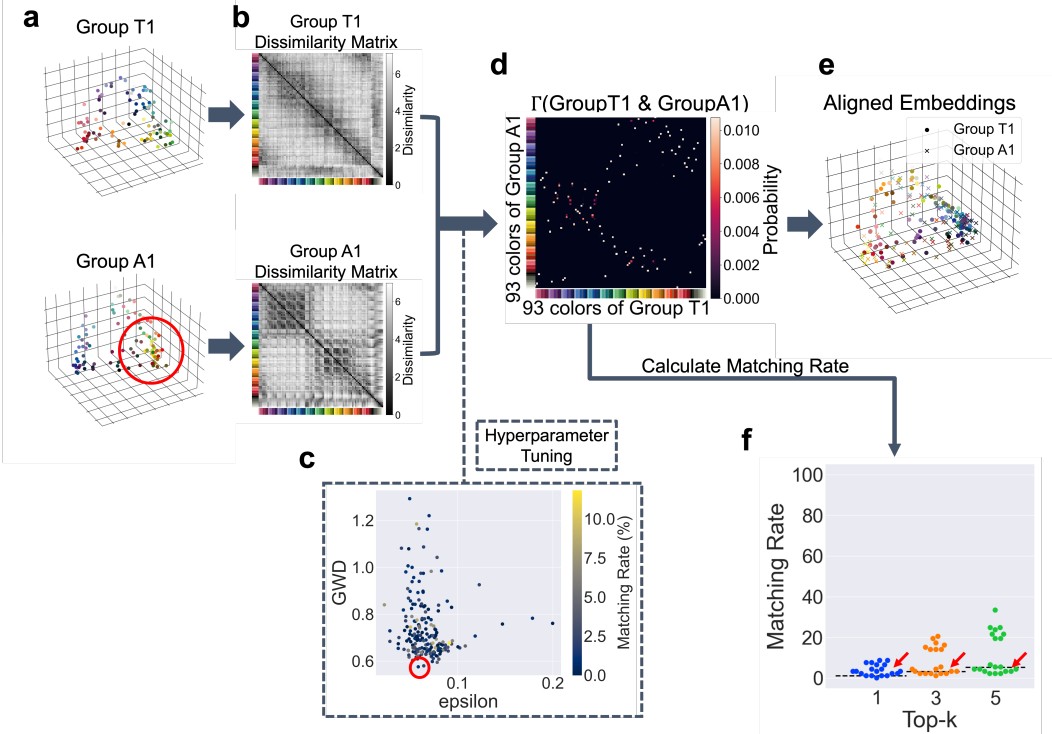

Figure 5: **Unsupervised alignment between qualia structures of color-neurotypical and atypical participant groups** (a) Estimated embeddings of 93 colors from one color-neurotypical participant group and one color-atypical participant group (Group T1 and A1). (b) Dissimilarity matrices of 93 colors from Group T1 and A1 obtained from the embeddings. (c) The optimization results over 200 iterations with different $\epsilon$ values. GWD values of local minima represented by points are shown with respect to $\epsilon$. Colors represent the top-1 matching rate of unsupervised alignment. (d) Optimal transportation plan $\Gamma$ between the dissimilarity matrices of Group T1 and A1. (e) Aligned embeddings of Group T1 and A1 plotted in the embedded space of Group T1. (f) The top-k matching rate of unsupervised alignment for 20 random pairs of participant groups. The chance levels are indicated by the dotted lines.

close to the respective chance levels (1.1%, 3.2%, 5.4%). Almost all values of the matching rates from 20 random samples are close to the chance levels, with a few exceptions with a matching rate above the chance level.

## 3    DISCUSSION

For a long time, assessing the similarity of subjective experiences across participants has been primarily considered as a philosophical question, rather than an empirical problem to be tackled scientifically (33; 34; 31; 32). While many previous studies have employed multidimensional scaling (MDS) to analyse structural aspects of dissimilarity judgments between perceptual experiences, they have invariably refrained from investigating whether qualia can be uniquely identified by their relational properties alone (18; 19; 20). To address this problem, we have proposed the "qualia structure" paradigm.

By using the proposed unsupervised alignment method, we were able to obtain qualitatively different results from those that would be obtained by the conventional supervised alignment method, such as representational similarity analysis (21). First, we showed that the qualia structures of colors within color-neurotypical or color-atypical participants can be aligned based only on the way the qualia are structurally related to each other, without using any external labels. One might think that these results are almost obvious, since the correlation between the dissimilarity matrices (Fig. 3b and Fig.

4b) are very high. However, in simulations, it is easy to create examples where the two structures are not correctly aligned, even when the correlation coefficient between two structures is very high (e.g., $\rho = 0.9$) (see Fig. 3 in (29)). In general, we cannot tell whether two structures are similar enough to align in unsupervised manner based on supervised measures of similarity such as the correlation coefficient alone.

In addition, we also showed that we could not unsupervisedly align the qualia structures of colors between color-neurotypical and color-atypical participants, even though the correlation coefficient between the dissimilarity matrices is reasonably high. Given the high correlation coefficient, the failure of the unsupervised alignment is not entirely expected. The unsupervised alignment probably failed because of the local structural difference, i.e., greenish colors and reddish colors are close in the embedding space of color-atypical participants (Fig. 5a), even though the overall structures are similar. Intriguingly, our results suggest that individuals with color-atypical vision may have a different structure of their color experiences, rather than just failing to experience a certain subset of colors. Longstanding thought experiments that challenge the feasibility of inter-subjective color comparisons, such as individuals with color qualia inversion (33; 34; 31; 35; 36), can be further constrained with our relational unsupervised approach. Beyond traditional measures such as Pearson's correlation coefficient, our method provides a more fundamental structural characterization of how two structures are similar or different, which will be crucial for future investigations of qualia structures across psychological, neuroscientific, and computational fields.

Although in this paper, we only considered group-based alignment because the number of trials obtained from each participant was insufficient for reliable unsupervised alignment (see Supplementary Fig. S2), it is interesting to consider individual-based alignment and assess the degree of the individual difference even for color-neurotypical participants. To obtain statistically reliable results, we would estimate from Supplementary Fig. S2 that at least more than 4,000 trials of similarity judgments are needed for each individual participant, which is almost same as the total number of pairs of 93 colors. We will conduct such experiments as future work.

While we focused only on color similarity, our method has the potential to be applied to a wide range of subjective experiences and different modalities (e.g. natural objects (37; 38), emotions (39; 40; 41), semantic concepts (30; 42), etc.). Our approach offers a novel and powerful tool for assessing the intersubjective correspondence of various qualia structures. If we rely only on our languages as the way to communicate our experiences, we are limited in understanding the consciousness of others. With the relational and structural approaches, we believe that we can construct an alternative and a more quantitative way to understand the sameness or difference in the consciousness experiences of others.

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

APPENDIX

EXPERIMENTAL DETAILS OF COLOR SIMILARITY JUDGMENTS

ETHICS

Experimental procedures were approved by the Monash University Human Research Ethics Committee (Project ID: 17674). Participants were provided electronically with written consent forms prior to the commencement of the experiment and provided electronic consent to participate. Participants were compensated for their time at a rate of £5.27 for an experimental duration of approximately 40 minutes.

DESIGN

**Participants**   Participants were recruited remotely through Prolific, an online participant recruitment platform. Participants accessed the experiment and provided data using their own personal computers. Only English native speakers were recruited. We recruited 488 general-population (color-neurotypical; (Group T)) and 548 self-identified color-atypical (Group A) participants prior to data cleaning.

**Exclusion - Color Typical**   Participants who failed to meet the inclusion criteria were excluded from the analysis. Firstly, we removed participants who failed to complete the experiment. Secondly, we excluded participants with a catch score (see below) of $< 77\%$. Catch trials were included to ensure participant attention and scattered randomly among the main trials. Lastly, the experiment was designed as a 'double-pass paradigm', meaning participants performed each sequence of main trials twice. Participants whose responses across the two passes were correlated $< 0.5$ were excluded, as low 'double-pass' correlation is indicative of inattentive or neglectful responding (16; 17). 62 out of 488 color-neurotypical participants were excluded, leaving 426 (87%) for the main analysis.

**Exclusion - Color Atypical**   We collected a cohort of 548 participants who self-identified as color blind. In addition to the general exclusion criteria, these participants were also screened using a modified online Ishihara test. Participants viewed a set of 28 Ishihara color plates and were asked to report the number they observed. 16 of the plates were standard and used as a positive control, with participants excluded if they correctly identified $> 80\%$ of the plates (i.e. made fewer than three errors) (43). 12 plates consisted of standard Ishihara plates that were red-shifted or blue-shifted so that the number should be correctly identifiable by participants with red-green color deficiencies (44)). These plates were used to detect participants who falsely identified as red-green color blind, with participants excluded if they correctly identified $< 80\%$. After these additional exclusion criteria, 257 of 548 (47%) participants who self-identified as color blind were used for the main analysis (Supplementary Fig. S1). As the vast majority of individuals with colour vision deficiencies possess some form of red-green colour blindness (deuteranomaly or deuteranopia), and because the Ishihara plates we selected did not screen for blue-yellow colour blindness (protanomaly or protanopia), it is probable that the included participants overwhelmingly possess some degree of red-green colour blindness (45). Our detailed analysis of the individual responses in Supplementary Fig. S1 supports this interpretation that our colorblind participants are largely uniform in their deficiency.

**Display apparatus**   Due to the nature of online experimentation, participants used their own computer screen to perform the experiment. The stimuli for the current study were based on the color swatches used by (46). This 93-color set was selected by (46) from the Practical color Co-ordinate System (PCCS). All stimuli were presented as solid colored circles 120 pixels in diameter on a grey (#7F7F7F) background.

**Procedure**   After recruitment through Prolific, participants were directed to the experiment hosted on Pavlovia. The first page of the experiment was a consent form that they could electronically sign by pressing the spacebar. Participants were informed that the data collection process was anonymous and that they could quit the experiment at any time. Following consent, participants were provided written instructions on how to complete the experiment. This was followed by 9 practice trials, seven of which were color similarity judgments and the rest were catch trials.

Main trials for color-neurotypical participants proceeded as follows. First, a fixation cross was presented in the centre of the screen for 250 ms. Following this, the two stimuli were presented as solid-colored circles for 250 ms. Considering the centre of the screen as the midpoint, each

stimulus was presented 180° apart and at a radius of 8% of the width of their screen. The stimuli were randomly assigned to a position within ±30° of horizontal meridian in order to prevent retinal adaptation between trials. Lastly, the participants were presented with a response screen and were directed to select a specified value from 0 (most similar) to 7 (most dissimilar). After responding, participants were asked to click on the centre of the screen to initiate the next trial. Atypical participants were presented with a slightly updated version of the same task. Instead of stimuli being presented randomly within ±30° of horizontal meridian, they were presented randomly in two out of four possible locations equidistant from the centre of the screen and maximally spaced from each other. Additionally, participants reported using values from $-4$ to $+4$ (with zero excluded) instead of 0 to 7. All other parameters remained the same.

Catch trials involved no presentation of colored stimuli patches. Instead, participants were shown a response screen where they were prompted to click a specific number. All other aspects of the response screen were the same.

During practice trials, participants were provided feedback on what selection they made, consisting of both the value they selected and the text 'Very Similar', 'Similar', 'Different' or 'Very Different' for selections of 0/1, 2/3, 4/5, 6/7 respectively for the color-neurotypical participants, or $-4/-3$, $-2/-1$, 1/2, 3/4 for the color-atypical participants. At the cessation of these practice trials they were asked to press the SPACE button to proceed to the main trial set. Following the practice trials, participants completed the main task. As with the practice trials, catch trials were randomly inserted among the main trials. Each participant was randomly allocated a set of color pairs out of the total 4371 unique pairs of 93 colors (including pairs of the same color), which were presented in a random sequence. Color-neurotypical participants were allocated 162 color pairs. After providing a response for each color pair once, color-neurotypical participants performed a repeat of the first 162 trials, identical in stimuli and sequence (double-pass). In total, this comprised of 324 main trials and 20 randomly interspersed catch trials. Color-atypical participants were allocated 81 color pairs, which were also presented in a double pass manner for a total of 162 main trials and 10 catch trials.

### ESTIMATION OF EMBEDDINGS AND DISSIMILARITY MATRICES AT THE GROUP LEVEL

**Aggregating similarity judgements**    To estimate embeddings at the group level, we aggregated similarity ratings from multiple participants. We fixed the number of similarity ratings taken from each participant to 75, which corresponds to the minimum number of unique color pairs judged among all color-neurotypical and color-atypical participants. We randomly chose 75 similarity ratings without replacement from each participant. Then, we aggregated the similarity responses from the fixed number of participants $Z$ and made a group of participants. The participants were chosen randomly from the entire participants (426 for color-typical participants, 257 for color-atypical participants).

To assess how many trials of similarity judgments are needed to reliably determine whether two similarity structures are aligned in an unsupervised manner, we varied the number of participants in a group, $Z = 16, 32, 64, 128$. As we can see in Supplementary Fig. S2, we found that $Z = 128$ (9600 trials) is necessary to obtain an accuracy of unsupervised alignment that is unquestionably higher than the chance level for any random samples. Based on this analysis, we only showed the results of the alignment when $Z = 128$ in the main text. See Supplementary Fig. S2 for the other cases.

**Estimation of embeddings based on the similarity judgements**    Based on the aggregated similarity judgement responses, we estimated the embeddings of 93 colors. The embeddings are estimated by training a one-layer linear neural network model with the similarity judgment data by using pytorch. The procedure is as follows.

First, the initial embedding of each color denoted by $e_i$ is given by a one-hot vector of 93 dimensions. Second, the initial embeddings are linearly transformed into 20 dimensional embeddings as

$$x_i = We_i, \tag{1}$$

where $x_i$ is the embedding of the $i$-th color, $W$ is the weights of the neural network that need to be learned so that the loss function defined below is minimized. We set the embeddings of dimensions large enough to capture the similarity structure of 93 colors. Note that to avoid over-fitting, the number of dimensions set here is effectively reduced by the hyperparameter tuning of the L1 regularization term through the usual cross-validation procedure.

The similarity ratings between the pair of colors are given by the Euclidean distance

$$D_{ij} = \|\boldsymbol{x}_i - \boldsymbol{x}_j\|^2. \tag{2}$$

Then, by using the empirically obtained similarity rating $S_{ij}$ for the color pair $i$ and $j$, the loss function to minimize is defined as

$$L = \sum_{(i,j)}^{n_{\text{train}}} \|D_{ij} - S_{ij}\|^2 + \lambda \sum_{i=1}^{m} \|\boldsymbol{x}_i\|_1. \tag{3}$$

where the summation of the first term is taken over all the color pairs $(i,j)$ in the training dataset, $n_{\text{train}}$ is the total number of color combinations in the training dataset, $m$ is the number of the colors, $\|\cdot\|_1$ denotes the L1 norm $\|\boldsymbol{z}\|_1 = \sum_i |z_i|$, and $\lambda$ is a hyperparameter that determines the strength of the $L1$ regularization. The loss function was optimized by the Adam algorithm with a fixed number of 100 epochs using pytorch. The hyperparameter $\lambda$ was optimized by 5-fold cross-validation.

UNSUPERVISED ALIGNMENT USING GROMOV-WASSERSTEIN DISTANCE

In this section, we provide an overview of unsupervised alignment methods for aligning two qualia structures (two sets of embeddings) by using Gromov-Wasserstein optimal transport. With this method, we can quantify the degree of similarity between the qualia structures. Also, the results from this analysis can inform us in what way those are similar or different, which can be examined by detailed analysis of the correspondence between the embeddings of the two qualia structures.

GENERAL PROBLEM SETTING

We consider the problem of aligning two sets of embeddings $X$ and $Y$, which in our case correspond to the embeddings of the color qualia structures. $X$ and $Y$ are $d \times n$ matrices where $n$ is the number of embeddings and $d$ is the dimension of embedding vectors.

$$X = \begin{pmatrix} \boldsymbol{x}_1 & \boldsymbol{x}_2 & \cdots & \boldsymbol{x}_n \end{pmatrix}, \quad Y = \begin{pmatrix} \boldsymbol{y}_1 & \boldsymbol{y}_2 & \cdots & \boldsymbol{y}_n \end{pmatrix}. \tag{4}$$

Here, $\boldsymbol{x}_i$ and $\boldsymbol{y}_i$ are column vectors, which are the embeddings of the $i$th-color quale of $X$ and $Y$, respectively.

The general problem setting in this study is to find the optimal alignment between $X$ and $Y$ without assuming any correspondence by solving the following problem:

$$\min_P \min_Q \|X - QYP\|_F^2, \tag{5}$$

where $\|\cdot\|_F$ is the Frobenius norm $\|A\|_F = \sqrt{\sum_{i,j} a_{ij}^2}$, $P$ is the $n \times n$ assignment matrix that establishes correspondence between the column vectors of $X$ and those of $Y$ (i.e., $\boldsymbol{x}_j \sum_i P_{ij} \boldsymbol{y}_i$), and $Q$ is the $d \times d$ orthogonal matrix that rotates $Y$ to fit into $X$. If we only allow one element in each column of $P$ to be 1 and set the other elements to 0, the problem becomes finding a one-to-one correspondence between the columns of $X$ and $Y$, or equivalently, finding the optimal permutation of the column indexes of $X$. In this study, we examine a more general scenario where the elements of matrix $P$ can take on a real number between 0 and 1. These values represent the degree of correspondence between the $i$-th column of matrix $X$ and the $j$-th column of matrix $Y$. This more flexible approach allows us to model the correspondences between the columns of $X$ and $Y$ in a more comprehensive manner.

SUPERVISED ALIGNMENT

When the assignment matrix $P$ is given, the optimization problem becomes the well-known Procrustes problem (47), which has a closed form solution. For instance, if we simply assume that the column indexes of $X$ match those of $Y$, and therefore $P$ is the identity matrix, the optimization problem is given by

$$\min_Q \|X - QY\|_F^2. \tag{6}$$

Given the singular value decomposition $U\Sigma V^\top$ of $XY^\top$, the solution to the Procrustes problem is given by $Q^* = UV^\top$.

UNSUPERVISED ALIGNMENT

In this study, we consider the scenario where the assignment matrix $P$ is not given. In this case, we need to jointly optimize $P$ and $Q$ in Eq. 5, which is a non-convex optimization problem without a closed-form solution. To address this, we first find an optimal assignment matrix $P$ using Gromov-Wasserstein optimal transport (GWOT) in an unsupervised manner. We then compute the Procrustes solution $Q^*$ based on the assignment matrix obtained from the GWOT analysis. This approach has been effective in unsupervised language translation tasks (25; 48). Denoting the optimal transportation plan (the assignment matrix) by $\Gamma^*$, the problem to solve becomes

$$\min_Q \|X - QY\Gamma^*\|_F^2. \tag{7}$$

The solution can be found by the singular value decomposition of $X(Y\Gamma^*)^\top$.

GROMOV-WASSERSTEIN OPTIMAL TRANSPORT

To obtain the assignment matrix $P$, which establishes the correspondence between the embeddings (the column vectors) of $X$ with the embeddings of $Y$, we use Gromov-Wasserstein optimal transport (GWOT) (23). GWOT is an unsupervised alignment technique that can find correspondence between two point clouds (embeddings) in different domains based on internal distances within each domain. Unlike classic optimal transport problems, the points in the two domains do not necessarily reside in the same metric space and any information about correspondences or distances between points "across" different domains is not given. In this study, the internal distances within the domains are represented by two different $n \times n$ dissimilarity matrices $D_{ij}$ and $D'_{ij}$ obtained from different participant groups, where $n$ is the number of colors and $D_{ij}$ denotes the subjective rating of dissimilarity between the $i$-th and $j$-th color.

The goal of Gromov-Wasserstein optimal transport problem is to find the optimal way to transport the distribution of resources (e.g., a pile of sand) from one domain to the other. There is a certain amount of the pile on each point in one domain. The distribution of the pile is given by $\boldsymbol{p}$ where $p_i$ is the amount of the pile at the $i$-th point in the source domain. We wish to transport the piles onto the points in the other domain so that the distribution of the pile matches with the target distribution $\boldsymbol{q}$ where $q_i$ is the amount of the pile at the $i$-th point in the target domain. With this setting, we wish to find the optimal transport plan that minimizes a certain transportation cost. The transportation cost considered in GWOT is given by

$$\min_\Gamma \sum_{i,j,k,l} (D_{ij} - D'_{kl})^2 \Gamma_{ik} \Gamma_{jl}. \tag{8}$$

Note that a transportation plan $\Gamma$ needs to satisfy the following constraints: $\sum_j \Gamma_{ij} = p_i$, $\sum_i \Gamma_{ij} = q_j$ and $\sum_{ij} \Gamma_{ij} = 1$. Under this constraint, the matrix $\Gamma$ is considered as a joint probability distribution with the marginal distributions being $\boldsymbol{p}$ and $\boldsymbol{q}$. We set $\boldsymbol{p}$ and $\boldsymbol{q}$ to be the uniform distributions, i.e., $p_i = q_i = 1/n$. Each entry $\Gamma_{ij}$ describes how much of the pile on the $i$-th point in the source domain should be transported onto the $j$-th point in the target domain. The entries of the normalized row $\frac{1}{p_i}\Gamma_{ij}$ can be interpreted as the probabilities that the embedding $\boldsymbol{x}_i$ corresponds to the embeddings $\boldsymbol{y}_j$.

With the transportation plan, the embeddings of $Y$ are mapped to the embeddings of $X$ as follows

$$\boldsymbol{x}_j \sum_{i=1}^n \Gamma_{ij} \boldsymbol{y}_i. \tag{9}$$

Then, this mapping is subsequently used for finding the rotation matrix $Q$ in Eq. 7.

HYPERPARAMETER TUNING

Previously, it has been demonstrated that adding an entropy-regularization term can improve the computational efficiency and help to find good local optimums of the Gromov-Wasserstein optimal transport problem (28; 24).

$$\min_\Gamma \sum_{i,j,k,l} (D_{ij} - D'_{kl})^2 \Gamma_{ik} \Gamma_{jl} + \epsilon H(\Gamma), \tag{10}$$

where $H(\Gamma)$ is the entropy of a transportation plan $\Gamma$ and $\epsilon$ is a hyperparameter that determines the strength of the entropy regularization.

To find good local optimums, we conducted hyperparameter tuning on $\epsilon$ in Eq. 10 by using the GWTune toolbox that we developed (29). This toolbox uses Optuna (49) for hyperparameter tuning and Python Optimal Transport (POT) (50) for GWOT optimization. We sampled 200 different values of $\epsilon$ ranging from 0.02 to 0.2 by a Bayesian sampler called TPE (Tree-structured Parzen Estimator) sampler (51). We chose the value of $\epsilon$, where the optimal transportation plan minimises the Gromov-Wasserstein distance without the entropy-regularisation term (Eq. 8) following the procedure proposed by a previous study (26).

INITIALIZATION OF TRANSPORTATION PLAN

To avoid getting stuck in bad local minima, it is effective to randomly initialize the transportation plan and try many random initialization, as proposed in (29). Each element in the initial transportation plan was sampled from the uniform distribution [0,1] and was normalized to satisfy the following conditions: $\sum_j \Gamma_{ij} = p_i$, $\sum_i \Gamma_{ij} = q_j$ and $\sum_{ij} \Gamma_{ij} = 1$. For each value of $\epsilon$, the transportation plan was randomly initialized.

EVALUATION OF UNSUPERVISED ALIGNMENT

To assess the degree of similarity between the two qualia structures in the unsupervised setting, the correct matching rate of color labels are computed between two groups based on the optimal transportation plan. Denote the color labels in group 1 and 2 as $c_1$ and $c_2$ respectively. The matching rate is calculated by comparing the transportation plan $\Gamma$ with these color labels. For each color $i$ in group 1, denoted by $c_{1i}$, the matching condition can be formalized as:

$$\text{Match}(i) = \begin{cases} 1, & \text{if } \Gamma_{ij} = \max_{j \in \{1,...,n\}}(\Gamma_{ij}) \text{ and } c_{1i} = c_{2j} \\ 0, & \text{otherwise} \end{cases} \tag{11}$$

This function indicates whether the $i$-th color in group 1, $c_{1i}$, matches with the same color in group 2, $c_{2j}$. The matching rate is then the percentage of colors in group 1 that match with the same colors in group 2, which can be calculated as

$$\text{Matching Rate} = \frac{\sum_{i=1}^{n} \text{Match}(i)}{n}, \tag{12}$$

where $n$ is the total number of colors ($n = 93$). In this study, the row and column of $\Gamma$ are sorted in the same order of colors and thus, the matching rate corresponds to the percentage of the diagonal elements $\Gamma_{ii}$ that are the largest among $\Gamma_{ij}$ for any $j$.

The matching rate defined above is top 1 matching rate. More generally, we also define top $k$ matching rate. For each color $i$ in group 1, we can define a function to determine if the probability of the $i$-th color corresponding to the same color in group 2 is within the top-$k$ probabilities:

$$\text{Top}_k(i) = \begin{cases} 1, & \text{if } \Gamma_{ij} \text{ is in the top-}k \text{ for } j \in \{1,...,n\} \text{ and } c_{1i} = c_{2j} \\ 0, & \text{otherwise} \end{cases} \tag{13}$$

The top-$k$ matching rate can then be calculated as

$$\text{Top-}k \text{ Matching Rate} = \frac{\sum_{i=1}^{n} \text{Top}_k(i)}{n}. \tag{14}$$

A high matching rate between two color similarity matrices suggests that two different groups have similar similarity structures of colors. Supplementary Figures S1, S2
Supplementary Movies S1, S2, S3

SUPPLEMENTARY FIGURES

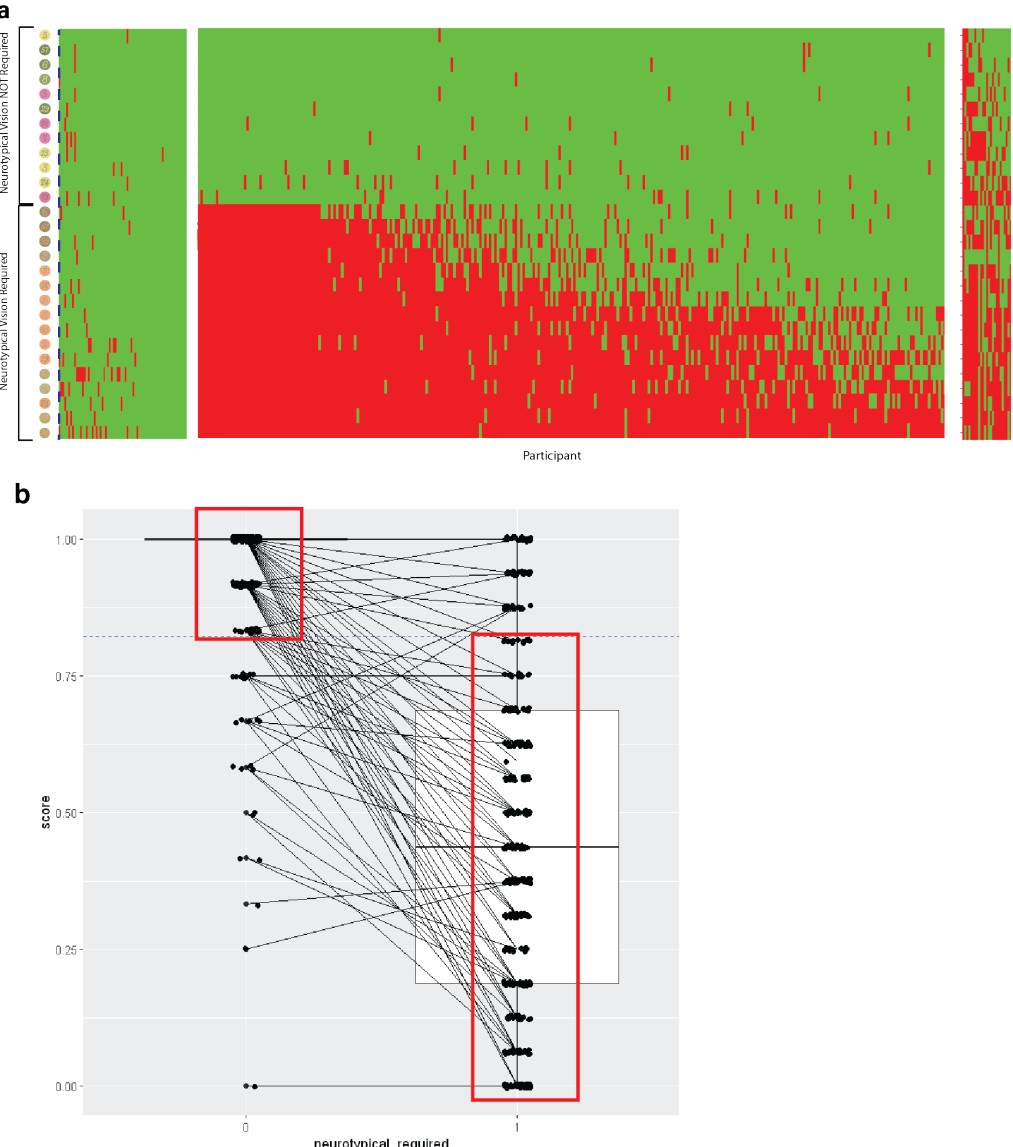

Figure S1: **Screening of participants with self-reported color blindness** (a) Each self-identified color-blind participant reported the number they observed for a set of 28 Ishihara color plates. 16 of the plates were standard and used as a positive control, while 12 plates were red- or blue-shifted so that even participants with red-green color vision deficiencies should still have been able to correctly identify the number visible (negative control). The performance of each participant for each plate is plotted in each column. Participants who scored greater than $80\%$ on the standard Ishihara plates were excluded (left). Participants who scored less than $80\%$ on the red- and blue-shifted Ishihara plates were excluded (right). All other participants were included, so long as they passed the rest of the exclusion criteria (middle). (b) Only participants who scored less than $80\%$ on the standard Ishihara plates, while scoring greater than or equal to $80\%$ on the red- and blue-shifted plates, and also passing the other exclusion criteria, were included in the main analysis.

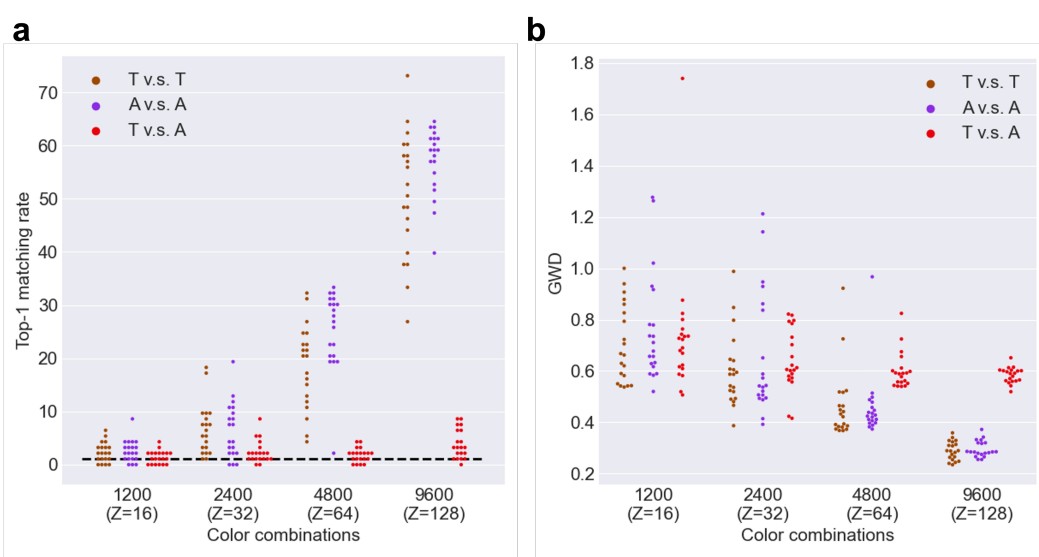

Figure S2: **Performance of unsupervised alignment with varying the number of trials for color similarity judgments.** (a) The top-1 matching rate and (b) GWD for color-neurotypical vs. color-neurotypical (T vs. T), represented by brown, color-atypical vs. color-atypical (A vs. A), represented by purple, and color-neurotypical vs. color-atypical (T vs. A) alignment, represented by red when the number of trials (or the number of participants $Z$) is varied. The chance level is indicated by the dotted line.

