# OpenReview forum: "Is my "red" your "red"?: Unsupervised alignment of qualia structures via optimal transport"
_ICLR.cc/2024/Workshop/Re-Align — ICLR 2024 Workshop Re-Align Poster_

### Official Review · Reviewer_RGgN · 2024-02-15
**Interesting results, solid paper but lack of engagement to the central theme of this workshop**

**Rating:** 1
**Fit:** 2
**Confidence:** 2

**Workshop Review:**

Summary:
This paper explores an important question of whether sensory experiences exhibit intersubjective equivalence, critiquing existing methods such as RSA for assuming supervised labels across individuals. Instead, the authors employ an unsupervised approach, revealing alignment within color-neurotypical and color-neuroatypical groups but misalignment between neurotypical and atypical groups.

Overall I think the paper is of interest to representational alignment research, well written and easy to follow. The experiments are designed logically and results presented clearly. However, a significant drawback preventing me from recommending its acceptance at ReAlign is its limited engagement to the central theme of this workshop, which is to bridge the gap in alignment research between biological and artificial systems. I think this paper made an excellent point that existing alignment approaches such as RSA assume matched intersubjective equivalence which is proven to be not the case in this paper. I find it fascinating given how widely adopted RSA is at evaluating representational alignment between neural activities in the brain and deep neural networks. Unfortunately, I couldn’t find any discussions on the implications of their findings to the current research on alignment between systems, which again, is the core of this workshop. This is a solid paper but I would not recommend its acceptance at ReAlign in its current form.

Below, I outline technical points that I would appreciate clarification on from the authors and presentation suggestions I hope they could consider for improving their manuscript:

Comments:
1. I am just curious if the conclusion is robust for methods other than GWOT? At least some discussion of the broader literature and what is expected could strengthen the message of the paper.

2. For Fig 5 results, I wonder if authors could identify if there is a subset of colors that particularly drives down the match rate. In other words, I’d assume these are colors whose perceptions are less affected by color blindness hence if considered separately would have led to a higher match rate.

3. In Discussion, the authors mentioned correlation coefficients were high between RDMs yet were unable to unsupervisedly align the qualia structures of colors between color-neurotypical and color-atypical participants. What was the actual correlation? I suggest you could even contrast the correlations with the match rate in the main results as it seems to be an important point that a popular approach like RSA could be misleading when the assumption of intersubjective equivalence doesn’t hold.

4. At the end of Discussion, the authors mentioned wider application to other modalities. It’s not entirely clear to me if this is the case for natural objects as opposed to color similarity. I’d imagine natural objects have much more robust labels across individuals than colors and perhaps there isn’t a good parallel to color blindness in object recognition?

**Reason For Not Giving Higher Score:**

I could not give a higher score due to the paper in its current form lacks clear engagement to the central topic of this workshop. While the research itself is very interesting and solid, relevance to the workshop needs to be greatly improved for me to raise the score.

**Reason For Not Giving Lower Score:**

The paper is overall well written, investigated an interesting question and has important implications for alignment research.

**Reviewer Domain:**

cognitive science

---

### Official Review · Reviewer_76hU · 2024-02-23
**creative approach to address an important question**

**Rating:** 3
**Fit:** 3
**Confidence:** 2

**Workshop Review:**

This work uses unsupervised optimal transport method for comparing "qualia structures" or the subjective experiences of color, aiming to answer if one person's perception of "red" is the same as another's. By analyzing subjective color similarity judgments from both color-neurotypical and color-blind participants without assuming prior correspondences between individuals' experiences, the study shows that color-neurotypical individuals' perceptions can be aligned, suggesting a structural equivalence in their color qualia. However, alignments between color-neurotypical and color-blind participants were unsuccessful, indicating a distinct difference in the subjective experience of color between these groups.

The paper offers a compelling method for exploring the subjective experience of color. It raises a question: if the study did not categorize participants as color-neurotypical versus color-blind and instead computed the alignment between any two individuals, would the results indicate two distinct populations or suggest a continuum? This query points to the potential for a nuanced understanding of color perception, exploring whether individual experiences form clear categories or if they exist along a gradient, blending the traditionally separate groups into a more complex spectrum of color perception.

**Reason For Not Giving Higher Score:**

recommended for talk

**Reason For Not Giving Lower Score:**

Creative approach for an important question

**Reviewer Domain:**

neuroscience

---

### Official Review · Reviewer_FpfA · 2024-02-27
**Interesting work comparing color-similarity ratings across groups of people (human vs. human comparisons).**

**Rating:** 2
**Fit:** 1
**Confidence:** 3

**Workshop Review:**

The authors introduce a novel method for comparing representational spaces across pairs of observers (in this case human vs. human judgments of color similarity). Although I think the work is very solid overall, I thought it would be helpful to clarify the added value of the approach over simpler methods (e.g., matching based on similarity scores alone).

Given the motivation for comparing individual observers (is my red your red?), I was surprised to see the analysis aggregated across subsets of 128 observers. Also, given than there were 426 participants total, why not just do random split halves? Why only 20 samples (rule of thumb is 500-1000, but that must depend on the variance and sample size…).

How many dimensions were the embedding spaces? How well does the embedding space for a group predict similarity of held-out pairs or held-out trials? In other words, how aligned are people with themselves given an estimated embedding space? In theory the alignment should be perfect if the embedding space was accurately estimated, so this self-consistency measure provides a useful “noise ceiling” (we can’t expect another group to align with group1 more than group1 aligns with itself).

I think it’s important to clarify how/why the outcome of the optimal-transport-aligned-embedding-space analysis differs from the results of directly analyzing the pairwise similarity ratings (e.g., after 2.2.1 Alignment between Color NeuroTypical observers). For example, what’s the top-1, top-3, top-5 matching rate using only the similarity ratings?

I understand why the authors have framed things in terms of qualia, given the philosophical motivations behind the project, but if I’m perfectly honest, the framing of this work in terms of qualia is a bit distracting, and could present a major obstacle for the broader audience. I worry that this work is deeply relevant to many people in the field who won’t really be able to get past this framing. This isn’t a critique of the work, just noting that as someone deeply interested in the empirical approach, I struggled with the emphasis on qualia.

Relatedly, I’m not really convinced that we can answer a question about the content of subjective experience by analyzing the alignment of the embedding spaces, so that conceptual issue made it hard for me to engage with the empirical approach independent of the qualia framing.

**Reason For Not Giving Higher Score:**

This work only compares humans to humans, on simple color similarity judgments, which seems outside the scope of the call (relating biological and artificial systems). If the method was also applied to model vs. human (and model vs. model) comparisons, I think it would be a great fit for the workshop (even more so if it went beyond color similarity judgments). It also wasn't clear to me how their method tells us anything we can't learn from the pairwise similarity ratings themselves (e.g., what's top-5 accuracy for predicting similarity ratings for 1 group from another group of participants? How could it be any lower than what you get from the embedding space?).

**Reason For Not Giving Lower Score:**

The methods are innovative, rigorous, and interesting if truly better than analyses based solely on the pairwise judgments themselves. I just wish they had applied the method to model vs. human comparisons, and used something higher-level than color judgments (though that's arguably high-level from the philosophical point of view; so really I just mean something that models are particularly good at, like comparisons between categories of objects).

**Reviewer Domain:**

cognitive science

---

### Decision · Program_Chairs · 2024-03-02

Accept (Poster)